# Forgotten Gems: Exploring the Untapped Benefits of Underutilized Legumes in Agriculture, Nutrition, and Environmental Sustainability

**DOI:** 10.3390/plants13091208

**Published:** 2024-04-26

**Authors:** Oluwatoyin A. Odeku, Queeneth A. Ogunniyi, Omonike O. Ogbole, Joerg Fettke

**Affiliations:** 1Department of Pharmaceutics and Industrial Pharmacy, University of Ibadan, Ibadan 200132, Nigeria; pejuodeku@yahoo.com; 2Department of Pharmacognosy, University of Ibadan, Ibadan 200132, Nigeria; queenethabiola@gmail.com (Q.A.O.); nikeoa@yahoo.com (O.O.O.); 3Biopolymer Analytics, Institute of Biochemistry and Biology, University of Potsdam, 14476 Potsdam, Germany

**Keywords:** underutilized legumes, agriculture, nutrition, environment, sustainability

## Abstract

In an era dominated by conventional agricultural practices, underutilized legumes termed “Forgotten Gems” represent a reservoir of untapped benefits with the unique opportunity to diversify agricultural landscapes and enhance global food systems. Underutilized crops are resistant to abiotic environmental conditions such as drought and adapt better to harsh soil and climatic conditions. Underutilized legumes are high in protein and secondary metabolites, highlighting their role in providing critical nutrients and correcting nutritional inadequacies. Their ability to increase dietary variety and food security emerges as a critical component of their importance. Compared to mainstream crops, underutilized legumes have been shown to reduce the environmental impact of climate change. Their capacity for nitrogen fixation and positive impact on soil health make them sustainable contributors to biodiversity conservation and environmental balance. This paper identifies challenges and proposes strategic solutions, showcasing the transformative impact of underutilized legumes on agriculture, nutrition, and sustainability. These “Forgotten Gems” should be recognized, integrated into mainstream agricultural practices, and celebrated for their potential to revolutionize global food production while promoting environmental sustainability.

## 1. Introduction

The term underutilized legumes refers to a diverse group of leguminous plants that have been historically overlooked or underused in mainstream agriculture and food systems, despite their potential nutritional, environmental, and socio-economic benefits [1]. These legumes encompass a wide range of species, many of which have been traditionally cultivated by indigenous communities or in specific regions but have received limited attention in terms of research, breeding, and commercialization compared to major crops like soybeans and common beans. The legumes, which include *Vigna radiata* (mung bean), *Macrotyloma uniflorum* (horse gram), *Psophocarpus tetragonolobus* (winged bean), and *Vigna subterranean* (Bambara groundnut), among others, possess genetic diversity and inherent benefits that have often gone unrecognized amidst the dominance of major legumes in global agriculture [2]. The primary reason why some legumes have not received the recognition they deserve and remain underutilized is an over-reliance on a few popular legume varieties in the global market. For example, common beans are widely consumed and cultivated, overshadowing lesser-known legume species [3]. Also, the lack of awareness and knowledge about the nutritional and health benefits associated with the consumption of a diverse range of underutilized legumes has led to limited demand and subsequently limited cultivation and distribution of these legumes [1]. However, there is a growing interest in promoting and exploring the potential of underutilized legumes. These legumes possess unique attributes that make them worth considering for sustainable agriculture and food security [4]. They are resilient to environmental stress and require little to no synthetic inputs, making them environmentally friendly and economically viable options for farmers [5]. Furthermore, underutilized legumes can potentially contribute to crop diversification and improve soil health [6]. They can be cultivated in rotation with other crops, thereby improving overall soil fertility and structure, and reducing the need for synthetic fertilizers [7]. Overall, underutilized legumes have immense untapped potential to improve food security, sustainable agriculture, and human health. By recognizing and valuing these lesser-known legume species, we can diversify our diets, enhance agricultural systems, and create a more resilient and sustainable food future. Despite the numerous potential benefits of underutilized legumes, their widespread adoption faces significant limitations, including low yield, agrotechnical challenges during cultivation, and low profitability of cultivation [8,9]. Through a comprehensive review of the existing literature, this report highlights the diverse benefits of underutilized legumes in agriculture, nutrition, and environmental sustainability, shedding light on their potential to address pressing global challenges, such as food security, malnutrition, and environmental degradation.

### 1.1. Benefits of Underutilized Legumes in Agriculture, Nutrition, and Sustainability

In the face of pressing challenges, such as food insecurity, malnutrition, and environmental sustainability, exploring the benefits of underutilized legumes in agriculture, nutrition, and sustainability has become increasingly significant (Figure 1) [10,11]. Legumes, a diverse family of plants, have long been recognized for their valuable contributions to these areas. While certain legume species have gained widespread attention and cultivation, many other underutilized legumes remain largely untapped. These underutilized legumes, with their unique characteristics and nutritional profiles, offer immense potential in addressing critical global issues of food security, malnutrition, and environmental sustainability [12]. Underutilized legumes play a critical role in enhancing agricultural productivity and resilience [13]. With their capacity to fix atmospheric nitrogen and improve soil fertility, these legumes offer sustainable alternatives to synthetic fertilizers. By incorporating underutilized legumes into cropping systems, farmers can reduce reliance on chemical inputs, prevent soil erosion, and foster sustainable farming practices [14,15]. Moreover, their adaptability to diverse soil conditions and ability to withstand drought make them invaluable assets for farmers operating in challenging environments [16]. Legumes are rich sources of essential nutrients, including proteins, vitamins, dietary fiber, minerals, and bioactive compounds [17]. By incorporating underutilized legumes into diets, especially in regions where protein-energy malnutrition is prevalent, the nutritional quality of meals can be significantly improved [18]. These legumes offer an affordable and sustainable source of high-quality protein, particularly in areas where animal-based protein sources are scarce or expensive [19]. Exploring the nutritional potential of underutilized legumes can contribute to combating malnutrition, improving public health, and reducing the burden of diet-related diseases [12]. Underutilized legumes play a vital role in promoting environmental sustainability. These legumes enhance soil fertility by means of their ability to fix atmospheric nitrogen, reducing the need for synthetic fertilizers that contribute to greenhouse gas emissions [20]. By incorporating underutilized legumes into agricultural systems, sustainable farming practices can be promoted, leading to reduced environmental impacts and increased resilience to climate change [21]. Nevertheless, despite their immense potential, underutilized legumes face various challenges that hinder their widespread adoption. Limited knowledge about their cultivation practices, lack of market demand, and limited research and development initiatives are significant barriers. Addressing these challenges requires collaborative efforts from researchers, policymakers, farmers, and consumers. These necessitate the development of farmer training programs, market development initiatives, and policy support to encourage the cultivation, processing, and consumption of underutilized legumes.

### 1.2. Rediscovering the Forgotten Gems

Underutilized legumes hold immense potential as forgotten gems, and they offer a variety of benefits that can contribute to food security, agricultural sustainability, and human well-being. These legumes are frequently rich in vital nutrients such as protein, vitamins, dietary fiber, and minerals [17]. They can provide a valuable source of nutrition, particularly in regions where malnutrition and micronutrient deficiencies are prevalent [22]. Promoting these legumes can help diversify diets and improve the nutritional status of communities where access to a variety of food sources is limited [8]. Underutilized legumes have the potential to adapt to various climatic conditions, including harsh environments characterized by poor soils, low rainfall, and high temperatures [23]. They possess unique traits, such as drought and heat resistance, as well as soil adaptability, making them resilient crops in the face of climate change. Their re-introduction into agricultural systems will help farmers enhance their resilience and reduce vulnerability to climate-related challenges. Underutilized legumes, like their more widely grown siblings, can fix atmospheric nitrogen via symbiotic interactions with nitrogen-fixing bacteria [24]. This process enriches the soil with nitrogen, reducing the need for synthetic fertilizers and improving soil health. This can help promote sustainable farming practices and reduce environmental degradation. These legumes contribute to the conservation of agricultural biodiversity [8]. Many of these legumes are indigenous to specific regions and possess genetic traits that can be valuable in crop improvement efforts. By promoting the cultivation of these forgotten gems, genetic diversity can be preserved. This is crucial for the development of disease-resistant varieties, adapting crops to changing environmental conditions, and ensuring long-term food security. In order to harness the benefits of underutilized legumes, it is essential to invest in research, development, and awareness initiatives to unlock their potential and reintegrate them into agricultural systems.

## 2. Agricultural Potential

### 2.1. Overview of Selected, Underutilized Legume Varieties

Underutilized legumes have considerable potential to contribute to sustainable agriculture and address several difficulties facing farmers and the food chain [23]. Unlike conventional legumes which extra measures will be taken to watch out for pests and diseases, underutilized legumes are highly resistant to pests and pathogens, they have also been said to strive in harsh conditions which makes special farming practices less of a concern for the farmers [6]. They also have a high potential for application in sustainable agricultural methods because of their capacity to replenish the soil [25]. They have found applications in shift cultivation, mixed farming, and crop rotation [23]. Other significant benefits of these legumes include their ability to fix atmospheric nitrogen through symbiotic relationships with nitrogen-fixing bacteria [26]. These legumes have distinctive genetic features and adaptations that make them resistant to the variability of environmental challenges, including heat, drought, and pests [27]. If given proper attention like conventional legumes, underutilized legumes including *Vigna radiata* (mung bean), *Macrotyloma uniflorum* (horse gram), and *Vigna subterranean* (Bambara groundnut), among other underutilized legumes, possess the ability to provide a sustainable supply of diverse nutritious food globally [28].

For instance, *Vigna radiata*, with the common name mungbean, a relatively important legume crop in South and Southeast Asia, has an annual production that has exceeded over six million hectares globally, which is relatively significant for an underutilized crop [29]. However, when compared to the areas covered by conventional legumes, mungbean production is still negligible [28]. Mungbean is a short-lived legume, and this makes it fit well into the fallow time between rice–wheat, rice–rice, rice–potato–wheat, maize–wheat, and other cash crops [30]. The cultivation of mungbean has been reported to help improve soil fertility and provide additional nitrogen to consequent crops [31]. Following the intercropping of mungbean with rice, the yield of rice was reported to increase by 8%, owing to the ability of mungbean to fix soil nitrogen and also reduce the effect of pests and disease [28].

*Macrotyloma uniflorum*, generally known as horse gram, accounts for roughly 5–10% of pulses output in India, with an annual yield of approximately 0.65 million tonnes [32]. Among underutilized crops, horse gram has an additional significance as a legume of indemnity to harsh environmental conditions [33]. It is a hardy legume with significant tolerance against different abiotic conditions, such as drought, heat, salinity, and heavy metal stress, with formidable pest resistance due to its antimicrobial properties [33]. Horse gram is primarily cultivated as an inter-crop or mixed crop with several cereals in all horse gram-growing areas [34]. It has been reported to have no adverse influence on the productivity and availability of fodder, except for minimizing undesired environmental impacts through reduced fertilizer and pesticide requirements [33]. It has also been documented for its usage in traditional mixed cropping to mitigate drought conditions, conserve crop diversity, and increase yield and profit [35].

*Vigna subterranean*, alternatively called Bambara groundnut, is an underutilized African legume that is grown throughout sub-Saharan Africa and in Southeast Asia, particularly, Thailand and Indonesia [10,13]. The legume is noted for its tolerance to drought and judicious yield, which is why it is still grown by local dwellers. It has been reported to exhibit all three drought tolerance strategies, which are avoidance, escape, and tolerance [36]. Bambara groundnut can resist a variety of climatic conditions and stress, from low temperatures at night and high temperatures during the day in a dry environment in Botswana to extremely temperate and humid circumstances in Indonesia [37]. Aside from the drought-resistant trait, Bambara groundnut has also been reported for its nitrogen-fixing capacity, thus contributing to soil fertility [10]. Due to resilience to adverse environmental circumstances and capacity to increase crop output, underutilized legumes are emerging as possible future food legumes that could significantly contribute to food and nutrition security globally.

### 2.2. Comparative Analysis of Commonly Cultivated Legumes with Underutilized Legumes

When comparing the potential of underutilized legumes with commonly cultivated legumes, several factors come into play, including nutritional value, genetic diversity, resource requirements, pest and disease resistance, environmental impact, cultural acceptance, market accessibility, potential for sustainable agriculture, and many more. Table 1 presents clear distinctions between these two categories of legumes. While the commonly cultivated legumes have the advantage of some of these factors, underutilized legumes also offer some unique characteristics.

### 2.3. Resilience and Adaptability of Underutilized Legumes to Diverse Climates and Soils

Similar to many crop species, underutilized legumes do respond to stress. They are embedded with an innate potential to survive in harsh environments with extreme drought, temperatures, pH, and salinity [6]. These particular adaptive processes allow them to be stress-resistant for survival and production even under harsh soil and agro-climatic conditions [38]. To avoid drought stress, legumes have developed certain effective strategies, including deep and prolific root systems, improved cell-wall flexibility to preserve tissue turgidity, osmotic adjustment, and strengthened antioxidant systems to protect cells from oxidative stress [39,40]. It is known that drought provokes various responses in plants which allows the plant to either avoid, escape, or tolerate drought [41]. Bambara groundnut, for instance, exhibits all three drought tolerance mechanisms (avoidance, escape, and tolerance). This makes the legume capable of tolerating a range of environmental conditions and durations of stress [42]. In response to projected climate change in South Africa, it has been projected that the yield and water productivity of bambara groundnut will increase by 37.5% and 33%, respectively [43]. It has also been demonstrated that, under climate change, the areas suitable for Bambara groundnut production would also expand in South Africa, confirming the resilience of the crop under climate change [13]. Additionally, the test color of Bambara groundnut has been used to predict drought resistance. Bambara groundnut genotypes with dark-colored seeds have been observed to have a greater emergence rate than their light-colored counterparts under stress circumstances, most likely due to the presence of tannins, which are known to act as antioxidants [13]. *Tylosema esculentum*, widely known as marama bean, has been documented for its unusual tuberous root that functions as a water and food reservoir, which permits it to resist extreme temperatures up to 50 °C in water-scarce circumstances and experience full dieback during the cooler months [44]. Apart from being impervious to various pests and diseases, horse gram has also been shown to survive water and salt stress and can be cultivated on marginal land with modest inputs due to its excellent antioxidant mechanisms, osmotic adjustment, and genetic and molecular processes [45].

## 3. Nutritional Value

### 3.1. Nutrient Content and Health Benefits of Underutilized Legumes

Similar to conventional legumes, underutilized legumes provide food and nutritional security to resource-poor rural populations during drought and extreme hunger, thereby, saving millions of lives [10]. They are frequently neglected, despite their high nutrient content and health advantages. Several underutilized legumes are extremely nutritious, capable of maintaining excellent health and guaranteeing food security [6] in a manner similar to that of the commonly cultivated legumes. Generally, legumes are a source of valuable nutrition, providing proteins ranging from 20 to 45% with some essential amino acids, dietary fiber ranging from 5 to 37%, complex carbohydrates of approximately 60%, vitamins, and minerals [8,27]. They have low cholesterol and are usually very low in fat with ±5% energy from fat, with the exception of peanuts (±45%), chickpeas (±15%), and soybeans (±47%) [46]. In addition to the nutritional components of legumes, they have been attributed to pharmacological roles due to the presence of important bioactive compounds. Studies have shown that phytochemicals found in underutilized legumes have antioxidant, anticancer, antidiabetic, antihypertensive, anti-inflammatory, antimicrobial, and hepatoprotective activities [47,48]. However, many people are unaware of the health advantages of their phytochemicals, therefore they have yet to be fully utilized for improved health and well-being. In Africa, these legumes have been reduced to the position of “poor man’s food”, with a very low degree of cultivation, production, consumption, and usage as compared to mainstream legumes [27]. Several underutilized legumes have been shown to have nutritional and health benefits. Some of these underutilized legumes and their nutritional constituents are listed in Table 2.

Bambara groundnut, a healthy underutilized legume, is grown locally in sub-Saharan Africa to sustain families in rural communities as a source of carbohydrates, protein, and other essential nutrients as well as income [10]. Its protein level has been determined to be comparable to other typical legumes and, hence, is regarded as a supplement for cereal-based diets [49]. Owing to the complex nutritional composition of the legume, it has been a research focus as a functional food alternative as well as nutraceuticals. Bambara groundnut cultivars from Zambia have been observed to have antioxidant activity equivalent to lentils, common beans, and chickpeas [50]. Flavonoids, tannins, and alkaloids have been discovered in the legume cultivars from Nigeria with the alkaloids reported to have analgesic properties [50,51]. Some other studies have also shown that Bambara groundnut possesses antimicrobial and antioxidant properties and has been used to treat many ailments [52,53]. The United States National Academy of Sciences has recognized horse gram as a possible food source for the future due to its robust nutritional components that are equivalent to other frequently grown pulses [54]. Horse gram proteins exhibit free radical scavenging activities which make them suitable as a food supplement and natural antioxidant [55]. Bioactive peptides of the legume’s protein have also been reported to possess antimicrobial, antioxidant, anticarcinogenic, and hypocholesterolemic activities [56].

**Table 2 plants-13-01208-t002:** Some underutilized legumes and their nutritional composition.

Underutilized Legume	Common Name	Nutritional Composition	Reference
*Vigna radiata*	Mung bean	Protein; whole seed and (17.9–25.3%), crude fiber (2.9–17.04%), carbohydrate (32.14–66.33%), crude fat (0.12%–2.31%), minerals, vitamins, and essential amino acids	[57]
*Macrotyloma uniflorum*	Horse gram	Protein; whole seeds (17.9–25.3%) and dehulled seeds (18.4–25.5%), carbohydrate; whole seeds (51.9–60.9%) and dehulled seeds (56.8 66.4%), essential amino acids, fat (0.6–2.6%), crude fiber (28.8%), minerals, and vitamins	[34]
*Vigna subterranean*	Bambara groundnut	Carbohydrates (57.43–63.09%), protein (18.0–24.0%), fiber (5.0–12%), minerals, and vitamins	[50]
*Tylosema esculentum*	Marama bean	Protein (29–38%), carbohydrate, dietary fiber (19–27%), lipids (32–42%), approximately 75% unsaturated fatty acids, minerals, and vitamins B6 and B12	[58]
*Sphenostylis stenocarpa*	African yam bean	Crude protein (24–28%), crude fiber (5.2–5.7%), carbohydrate (74.10%), fat (1.5–2.0%), some minerals, and vitamins	[59]
*Lablab purpureus*	Lablab	Crude protein (20.46–25.47%), carbohydrate (60.63–66.32%), crude lipid (2.69–4.17%), and dietary fiber (4.98–6.90%)	[60]
*Psophocarpus tetragonolobus*	Winged bean	Protein (32–37%), carbohydrate (23–40%), fat (14–25%), minerals, and vitamins B1, B2, B6, B9, A, C and E.	[61]
*Macrotyloma geocarpum*	Kersting’s groundnut	Protein (21.3%), carbohydrate (61.53–73.3%), fiber (6.2%), and amino acids including arginine, phenylalanine, and histidine	[62]

### 3.2. Contribution of Underutilized Legumes to Dietary Diversity, Malnutrition, and Food Security Issues

Persistently high levels of malnutrition may be linked to less dietary diversification and many national food-based dietary guidelines uphold the notion of dietary diversity [63]. This is most likely because it is a cost-effective, inexpensive, and sustainable way to alleviate hunger and malnutrition while delivering enough or optimum nutrition [64]. The incorporation of underutilized legumes into the diet has the potential to promote dietary diversity while decreasing the prevalence of malnutrition. They contribute to the global fight against hunger and malnutrition, which is typically caused by insufficient intake of proteins and other essential vitamins and minerals [64]. Underutilized legumes are essential for boosting dietary and production diversity [65]. They provide vital nutrients, including proteins, lipids, fiber, carbohydrates, amino acids, fatty acids, vitamins, and minerals, in the appropriate proportion, with adequate diversity for healthy growth, while also reducing the risk of diet-related disorders [12]. These legumes are an appropriate option for persons aiming to avoid or minimize their consumption of animal products, which have been related to the development of disorders such as obesity and diabetes [1]. Food insecurity and malnutrition are a global concern, worsened mainly by the expanding population and diminishing crop productivity. Food security is endangered by human overdependence on less than 1% of edible plant species [66,67]. Malnutrition caused by protein deficiency has been linked to a high mortality rate, greater costs for disease treatment, and a low rate of recovery from illness [68]. It has also been reported to be responsible for 45% of all deaths in children worldwide [69]. In order to improve food security and address the issue of malnutrition, emphasis has been placed on legumes. While a great focus has been placed on conventional legumes over the years, some underutilized legumes are beginning to receive major recognition as a potential answer to alleviate malnutrition and increase food security [70]. Underutilized legumes with high nutritional value and good climate tolerance, such as marama bean and Bambara groundnut, can help overcome these issues and can potentially provide sustainable food resources in the future [71]. Like many other underutilized legumes, Marama bean and Bambara groundnut are a rich source of protein, one of the essential macronutrients essential for optimum human growth and development [58,72]. They can assist in alleviating the global protein shortage in the future by partially or completely replacing animal proteins in the human diet. However, despite the obvious promise of underutilized legumes for sustainable agriculture, underutilized legumes like Marama bean only grow in the wild, whereas Bambara groundnut is produced by small-scale farmers as a survival crop [71]. To fully realize the promise of underutilized legumes as a sustainable protein source, efforts must be made to mainstream their cultivation and domestication. This will assist in combatting protein deficiency and ensure global food security.

## 4. Environmental Sustainability

### 4.1. Underutilized Legumes with Nitrogen Fixation and Soil Health Improvement Potentials

The depletion of nitrogen in soil in various parts of the world has now developed into a serious concern for food production and security [16]. Several approaches, notably the use of chemical fertilizers, have been developed in recent years to address this problem, but their usage has been impeded by variables, such as environmental pollution, health issues, and the negative impact of climate change [6]. Improved substitutes for replacing depleted nitrogen in soil have been investigated, including the use of biological nitrogen fixation (BNF) with legumes. Nitrogen fixation by legumes is bacteria-mediated [25]. Legumes fix nitrogen in soil by symbiotically associating and cooperating with the rhizobia present in their root nodules to create ammonia utilizing the nitrogenase enzyme [6,73]. They have been estimated to fix up to 200 kg N_2_ per hectare provided that the soil is well-drained with a temperature of not less than 7 °C [74]. Nitrogen fixation potentials have been reported in underutilized legumes such as *Macrotyloma geocarpum* (Kersting’s groundnut), *Vigna subterranean* (Bambara groundnut), *Sphenostylis stenocarpa* (African yam bean), and *Psophocarpus tetragonolobus* (winged bean) [75]. The nitrogen-fixing ability of Kersting’s groundnut was reported to be expressed by *Bradyrhizobium*
*species*, *Lablab purpureus* (lablab), *Mesorhizobium ensifer*, and *Bradyrhizobium*
*species* [26,76]. *Macrotyloma geocarpum* (Kresting’s groundnut), fixed 16.5–57.8 kg N ha while lablab fixed 43–53 kg N_2_ per hectare. Bambara groundnut has also been reported to fix approximately 4–200 kg N_2_ per hectare via its symbiotic relationship with the soil bacteria, rhizobia [77]. The symbiotic connection of these underutilized legumes with these bacteria was found to improve plant growth, leaf chlorophyll, nodulation, and photosynthesis, resulting in improved output. Furthermore, the nitrogen-fixing potential of legumes frequently contributes to the enhancement of soil health because of their ability to boost soil nitrogen content [25]. Having known legumes to be cover crops and important in soil conservation and health, an increase in the total nitrogen of the soil left to fallow after cropping some legumes, including soybean, cowpea, lablab, and groundnut, was observed [78]. This has been reported in the rotation cultivation of Bambara groundnut with upland crops [79]. Underutilized legumes have great potential to boost nitrogen supply in soil and the overall health of the soil, both of which have a positive impact on the yield of the crop.

### 4.2. Relevance of Biodiversity Conservation of Underutilized Legumes

Biodiversity is critical to mitigating the burden of malnutrition as the world’s population is anticipated to exceed 10 billion by 2050 [80]. This has led to a better awareness of the importance of biodiversity in ensuring food and nutrition security. As a result, the advantages of adopting underused legumes and investigating their sustainable production methods are essential. Including underutilized legumes in mainstream agriculture can improve resilience and provide sustainable food production systems. As global biodiversity is rapidly diminishing and restricting the possibility of discovering new food sources, an assessment of the current distribution and conservation status of underutilized legumes to support science-based decision-making is vital [81]. Furthermore, the negative consequences of climate change on biodiversity, agricultural output, and food security have made the preservation of food diversity and related traditional knowledge a global priority [82]. Additionally, many underutilized legumes, like other underutilized species, are at risk of extinction owing to shifting cultural views and lack of documentation [81]. Therefore, preserving underutilized legumes will not only conserve genetic diversity but also protect traditional agricultural practices, cultural heritage, and local food systems. Integrating traditional knowledge with modern conservation efforts can lead to more effective biodiversity conservation strategies that are socially and culturally appropriate. Conservation efforts for underutilized legumes may involve a combination of in situ and ex situ conservation strategies, where in situ conservation focuses on preserving underutilized legumes within their natural habitats, such as through the establishment of protected areas, on-farm conservation, and community-based seed banks. Ex situ conservation involves the collection, preservation, and maintenance of genetic resources outside their natural habitats, such as in gene banks and botanical gardens [83]. Both approaches are essential for safeguarding the genetic diversity of underutilized legumes and ensuring their long-term conservation.

## 5. Challenges and Solutions

### 5.1. Identification of Obstacles in the Widespread Adoption of Underutilized Legumes

Despite the immense potential of underutilized legumes, they are still faced with various challenges that hinder their adoption. One of the primary obstacles to the widespread adoption of underutilized legumes is the lack of awareness and knowledge among farmers, consumers, and even researchers [84]. Many people are not familiar with underutilized legumes, their nutritional benefits, or their culinary versatility [8]. This lack of awareness leads to limited demand and investment in research, breeding, and development efforts for underutilized legumes. Some underutilized legumes may have low yield potential or may require specific management practices that farmers are unfamiliar with, making it difficult for them to adopt and cultivate on a large scale [8]. Addressing this obstacle requires a multi-faceted approach with a focus on targeted breeding programs and improved agronomic practices. Commonly cultivated legumes, such as *Glycine max* (soybean) and *Phaseolus vulgaris* L. (common beans), have undergone extensive breeding programs aimed at improving yield potential and agronomic traits. These efforts have resulted in the development of high-yielding varieties that are well-adapted to various environmental conditions. In contrast, many underutilized legumes still lack the same level of genetic diversity and breeding attention. As a result, their varieties may exhibit lower yield potential and less desirable agronomic traits [85].

For instance, the production of Bambara groundnut in Africa has been recorded to be approximately 0.3 million tonnes annually. Although the yield of this legume in Africa varies between landraces and locations (0.5–3 t ha^−1^) [86], the average yield of 0.85 t ha^−1^ was reported to be comparable to other legumes [87]. In spite of the favorable characteristics of Bambara groundnut, farmers in sub-Saharan Africa still obtain low yields. Hence, the use of sophisticated breeding programs employing cutting-edge tools, distinct from traditional or conventional methods, has the potential to significantly advance the development of desired traits, ultimately leading to the creation of high-yielding varieties of underutilized legumes [88]. The market dynamics surrounding underutilized legumes can be a significant obstacle to their widespread adoption. Unlike underutilized legumes, commonly cultivated legumes, such as beans, lentils, and chickpeas, have well-established markets and value chains [9]. They are widely traded, consumed, and integrated into various food products. On the other hand, underutilized legumes are faced with limited market demand without well-developed distribution channels. This lack of market demand makes it challenging for farmers to find buyers for their underutilized legume crops [89]. Policy and infrastructure limitations can impede the widespread adoption of underutilized legumes [90]. In some cases, government policies may favor the cultivation and promotion of certain crops over underutilized legumes, leading to limited support and incentives for farmers to grow them [21]. Additionally, the lack of appropriate infrastructure, such as storage facilities, processing units, and market linkages, can make it difficult for farmers to handle and market underutilized legume crops effectively [90]. Furthermore, underutilized legumes often receive limited attention and investment in research and development [8]. This challenge hinders the improvement of underutilized legume varieties in terms of yield potential, disease resistance, and agronomic traits. Without adequate breeding programs and research initiatives, it becomes challenging to overcome the agronomic challenges and enhance the overall potential of underutilized legumes [91].

### 5.2. Solutions and Strategies for Overcoming Challenges

Overcoming the obstacles in the widespread adoption of underutilized legumes requires a multi-faceted approach (Figure 2). This includes increasing awareness and knowledge about the benefits of underutilized legumes [8]. Educational campaigns targeting farmers, consumers, and policymakers can help promote the understanding and appreciation of these legumes. This can be achieved through workshops, training programs, demonstration plots, and the dissemination of educational materials [92]. Investing in research on underutilized legumes by conducting studies to improve their productivity, adaptability to different agro-climatic conditions, and nutritional qualities may also help overcome the challenge of the widespread of forgotten gems [93].

Additionally, collaborative research efforts involving agricultural institutions, universities, and farmers can help generate valuable knowledge and technologies for the cultivation and utilization of underutilized legumes [94]. Enhancing the availability and quality of underutilized legume seeds is another critical approach. This can be achieved through the establishment of seed production and distribution networks specifically for underutilized legumes. Collaboration between seed companies, agricultural institutions, and farmers’ organizations can help ensure the availability of diverse and high-quality seeds to farmers [95]. Also, developing infrastructure to support the cultivation, processing, and marketing of underutilized legumes by investing in processing facilities, storage infrastructure, and transportation networks is essential [9]. This will help improve post-harvest handling and processing capabilities, thereby, enhancing the value addition and marketability of these legumes. Furthermore, creating market linkages and value chains for underutilized legumes by connecting farmers with potential buyers, such as retailers, food processors, and exporters will also contribute to the adoption of these legumes. By developing market channels, promoting product diversification, and supporting value-added processing, an increase in the demand and profitability of underutilized legumes can be achieved [96]. Ultimately, policy and regulatory reforms are also necessary to support the adoption of underutilized legumes. Governments can provide incentives and support mechanisms for farmers engaged in underutilized legume production. Policies that promote research and development, seed systems, infrastructure development, and market access for underutilized legumes can help create an enabling environment for their adoption [8]. By implementing these solutions, stakeholders can work together to overcome the obstacles and promote the widespread adoption of underutilized legumes. This will contribute to sustainable agriculture, improve nutrition, and resilient food systems.

### 5.3. Importance of Research, Awareness, and Policy Support

Research, awareness, and policy support play significant roles in overcoming the obstacles in the widespread adoption of underutilized legumes. Through research, scientists can identify suitable varieties, develop improved agronomic practices, and address challenges related to pests, diseases, and climate resilience [27]. Research also helps in enhancing the nutritional qualities and value-added processing techniques of underutilized legumes. Investing in research, scientific knowledge and evidence to support the adoption and promotion of these legumes can improve the utilization of these legumes [94]. Awareness creation about underutilized legumes is key to their widespread adoption. Farmers, consumers, and policymakers need to be informed about the nutritional benefits, environmental advantages, and economic opportunities associated with these legumes. Awareness campaigns can help dispel misconceptions, highlight success stories, and showcase the potential of underutilized legumes [92]. By raising awareness, interest in underutilized legumes can be developed by both farmers and consumers.

Policy support is essential for creating an enabling environment for the widespread adoption of underutilized legumes. Governments can play a crucial role by implementing policies that incentivize farmers to cultivate these legumes, support research and development, and establish market linkages [97]. Policies can also address regulatory barriers, promote seed systems, and provide financial support to farmers engaged in underutilized legume production [95]. With the integration of research, awareness, and policy support, the full potential of underutilized legumes can be unlocked.

## 6. Future Prospects

### 6.1. Potential for Further Research and Innovation

Underutilized legumes offer great potential for the development of value-added products and novel food formulations. For instance, legume flours, protein isolates, and functional ingredients can be used to create gluten-free products, meat substitutes, and fortified foods [98]. Additionally, the by-products of legume processing, such as husks and shells, can be utilized for biofuel production, animal feed, or as a source of bioactive compounds [99]. Research efforts can also focus on improving the post-harvest handling and processing techniques for underutilized legumes. This can include investigating optimal storage conditions, reducing post-harvest losses, and developing efficient processing methods to enhance their marketability and shelf life [21]. Moreover, exploring the genetic diversity within underutilized legumes can reveal valuable traits that can be utilized in breeding programs. By identifying and characterizing the genes responsible for desirable traits, such as disease resistance or high yield, researchers can accelerate the development of improved varieties [100]. Investing in the exploration of underutilized legumes can help uncover new sources of nutrition, create innovative food products, and contribute to sustainable agriculture [70]. To achieve these goals, it is crucial to recognize the value of these underutilized legumes and support efforts to promote their cultivation, research, and utilization.

### 6.2. Integration into Mainstream Agricultural Practices

Underutilized legumes have the potential to enhance the sustainability and resilience of agricultural systems [101]. As climate change poses increasing challenges to agricultural productivity, the integration of these legumes can contribute to climate change adaptation. Their ability to thrive in diverse environmental conditions can help farmers mitigate the impacts of climate change and maintain food production in the face of changing climatic patterns [102]. Also, as global populations continue to grow and face challenges of malnutrition and food insecurity, the integration of underutilized legumes will help provide nutritious food sources, especially in regions where access to animal-based protein is limited [84]. Additionally, as consumer demand for sustainable and nutritious food increases, there is potential for these legumes to become sought-after ingredients in the food industry [103]. Value-added products, such as legume flours, protein isolates, and functional ingredients, can cater to the growing market for plant-based, gluten-free, and allergen-free food products [104]. The integration of underutilized legumes into mainstream agriculture will contribute to the conservation of unique genetic resources and prevent the loss of valuable legume species [18]. This conservation is crucial for maintaining resilient agricultural systems and adapting to changing environmental conditions. Furthermore, the development of value-added products from underutilized legumes and the establishment of supply chains can generate employment opportunities, particularly for small-scale farmers and women in rural communities [105]. To realize these prospects and maximize the benefits of underutilized legumes, continued research, innovation, and collaboration among stakeholders are essential.

### 6.3. Implications for Global Food Systems and Sustainability Goals

According to estimates from the Food and Agriculture Organization (FAO) of the United Nations, 821 million people globally faced food and nutrition insecurity in 2017, with developing countries having the greatest rate of hunger [106]. Eliminating this issue is one of the Sustainable Development Goals (SDGs) of the 2030 agenda, which was endorsed by the United Nations, with a focus on developing and less-developed countries [107]. One strategic method to tackle this issue is to encourage biodiversity and the use of underutilized and neglected crops. Underutilized legumes can contribute to achieving zero hunger by boosting food production levels, diversifying the human diet, and ultimately, improving the sustainable use of a broad range of climate-smart legumes [1]. Additionally, owing to the nutritional and health benefits underutilized legumes offer, they are capable of addressing malnutrition and micronutrient deficiencies as well as preventing some chronic diseases [23]. Underutilized legumes can potentially contribute significantly to several other SDGs. They can significantly increase soil health and food production, both of which are beneficial for coping with the consequences of climate change and other environmental problems [108]. Moreover, the cultivation and consumption of these legumes can create income-generating opportunities for small-scale farmers, especially in rural areas because they can be grown with low inputs, making them accessible to resource-limited farmers and contributing to poverty reduction [21].

## 7. Conclusions

Exploring the benefits of underutilized legumes is of utmost significance in addressing global challenges. These legumes have the potential to enhance agricultural productivity, improve nutrition, and contribute to environmental sustainability. However, this exploration is not without limitations. Limited research on underutilized legumes hinders a comprehensive understanding of their potential benefits, including their nutritional content, agronomic requirements, and environmental impacts. Additionally, their limited genetic diversity poses challenges in breeding resilient varieties. Market demand for underutilized legumes is uncertain, and cultural preferences may hinder their adoption. Specialized processing techniques may be needed to make these legumes palatable and digestible, requiring investment in infrastructure. Policy and investment priorities often favor major crops, leaving underutilized legumes with limited support. Environmental considerations, such as water usage and land efficiency, need to be carefully assessed. Social equity must also be addressed to ensure equitable access to resources and support for smallholder farmers. Despite these limitations, promoting the exploration of underutilized legumes is essential for achieving global food security, improving public health, and mitigating the environmental impacts of agriculture. Stakeholders must collaborate to overcome these challenges and unlock the full potential of underutilized legumes, paving the way for a sustainable and resilient future.

## Figures and Tables

**Figure 1 plants-13-01208-f001:**
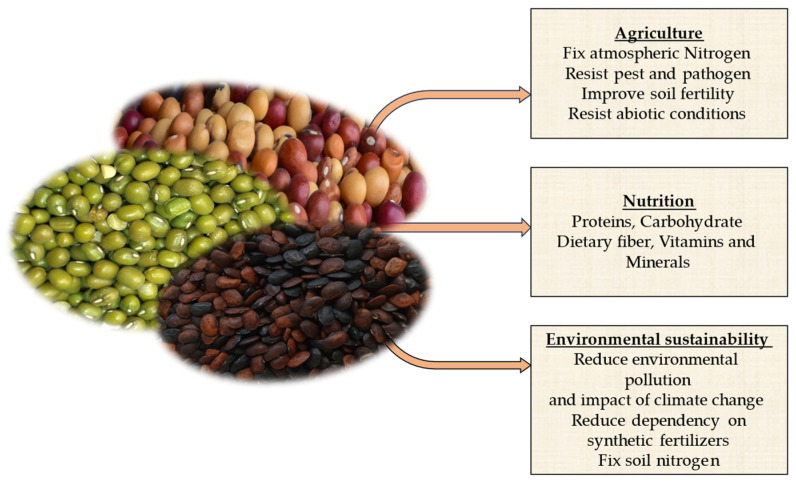
Roles of underutilized legumes in agriculture, nutrition, and environmental sustainability towards enhancing global food security.

**Figure 2 plants-13-01208-f002:**
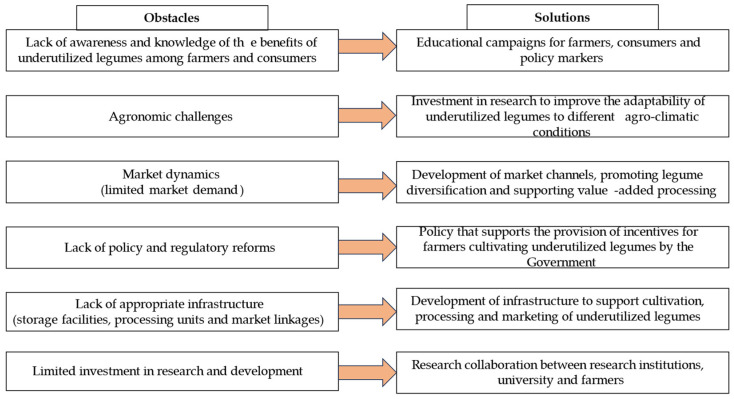
Obstacles hindering the widespread adoption of underutilized legumes and their solutions.

**Table 1 plants-13-01208-t001:** Distinctions between commonly cultivated legumes and underutilized legumes.

Aspect	Commonly Cultivated Legumes	Underutilized Legumes
Nutritional value	Well-studied and consistent nutritional profiles	Generally high, but may vary widely
Genetic diversity	Extensively bred with higher genetic diversity	Often limited due to less breeding
Environmental impact	Mixed, depending on farming practices	Generally lower ecological footprint
Market accessibility	Well-established markets and distribution channels	Limited market access and awareness
Potential for sustainable agriculture	Generally supportive of sustainable practices	Potential to enhance soil health and biodiversity
Adaptability	Often widely adaptable to various climates and soil types	May have niche adaptability to specific environments
Pest and disease resistance	Often bred for resistance to common pests and diseases	Can tolerate specific pests and diseases
Yield potential	Generally higher due to extensive breeding and optimization	Variable, often lower due to limited breeding and optimization
Soil health	Can vary, but may require additional inputs for soil health	Generally beneficial due to nitrogen fixation properties
Crop rotation benefits	Can provide benefits but may not be as pronounced	Often advantageous due to nitrogen-fixing capabilities
Input requirements	Often require more inputs for optimal yield	May require fewer inputs such as fertilizers due to nitrogen fixation
Cultural acceptance	Widely accepted and integrated into diets	Varied, may be less widely accepted

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
