# Peer review of "Forgotten Gems: Exploring the Untapped Benefits of Underutilized Legumes in Agriculture, Nutrition, and Environmental Sustainability"

_plants, 2024, doi:10.3390/plants13091208_

Round 1

Reviewer 1 Report

Comments and Suggestions for Authors

The paper presents the opportunity to use underutilized legumes.

Abstract: it is well written.

Introduction: the novelty of the study is not highlighted. The aim is not mentioned at the end of this section.

Section 2.2. Some numerical data would better support this chapter. A table or infographic can be made.

Section 3.1 I strongly recommend adding a table with numerical data, a comparison between underutilized legumes and modern ones for example. 

Overall, the review needs to be completed with some specific numerical data to better support the information exposed.

Conclusion: Please add the limitations of the study.

Comments on the Quality of English Language

English is fine.

Author Response

Manuscript ID plants-2936520

Title: Forgotten Gems: Exploring the Untapped Benefits of Underutilized Legumes in Agriculture, Nutrition, and Environmental Sustainability

Dear Reviewer we thank you for your valuable work and the suggestions. The comments have been handled and a point-by-point explanation has been provided.

Reviewer 1

The paper presents the opportunity to use underutilized legumes.

Abstract: it is well written.

Introduction: the novelty of the study is not highlighted. The aim is not mentioned at the end of this section.

Response: The novelty has been highlighted and the aim has been mentioned.

Section 2.2. Some numerical data would better support this chapter. A table or infographic can be made.

Response: A table highlighting the distinctions between commonly cultivated legumes and underutilized legumes has been provided.

Section 3.1 I strongly recommend adding a table with numerical data, a comparison between underutilized legumes and modern ones for example. 

Response: A table with the numerical data on the nutritional content of some underutilized legumes has been provided in Table 2 while general information on the nutritional content of commonly cultivated legumes has been provided in 3.1

Overall, the review needs to be completed with some specific numerical data to better support the information exposed.

Response: Numerical data has been provided where necessary.

Conclusion: Please add the limitations of the study.

Response: The limitation of the study has been provided.

Reviewer 2 Report

Comments and Suggestions for Authors

The article is interesting and contains important information about sustainable agriculture and nutrition. Its main weakness is the lack of specific examples or case studies that discuss the concepts presented.

The 'Introduction' completely lacks a clear definition of "underutilized legumes" and examples of such species. The weaknesses of these plants, such as low yields, agrotechnical problems during cultivation and low profitability of cultivation, were also completely ignored.

Chapter 1.1 should have the shortened title: Benefits of underutilized legumes in agriculture, nutrition and sustainability.

Line 79-80 - I do not agree with the statement that legumes have lower water needs than many other crops, which is why they can be grown in water-poor regions. Please provide literature data confirming this statement.

In chapter 1.2 (line 104-105) it is written that these plants have unique features, such as resistance to drought and heat, which makes them resistant to climate change, and in chapter 5 (line 381-383) and elsewhere, there is a complete contradiction of this.

The title of chapter 2.1 needs to be edited and shortened. Moreover, the review of "underutilized legumes" is very superficial and covers only 3 species. I suggest supplementing and expanding this chapter.

Similar comments apply to chapters 2.3 and 4.3. They require supplementation and more extensive editing.

Lines 221-223 - I disagree with the statement "Anti-nutrients in underutilized legumes including, α-amylase inhibitors, phytic acids, and lectins have been linked to their ability to resist pests and diseases”. Anti-nutritional substances are also found in popular legumes, which does not make them resistant to diseases.

The nomenclature should be unified - species names in Latin (see lines 255, 350 et al.)

Table 1 - should contain specific numerical data on the content of individual nutrients. The fact that legumes contain protein and carbohydrates is not a discovery.

Chapter 3.2 and 3.3 should be combined.

Chapter 4.1. is a repetition of what was written earlier. It doesn't show anything new.

Chapter 4.2. provides very general information. There is a lack of numerical data from the literature on the amount of nitrogen fixation by specific legume species.

Chapter 5.1 - the weaknesses of "underutilized legumes" related to low yield potential are presented too briefly and superficially. There is no comparison with other legume species that are popular and widely cultivated.

"Conclusions" require extensive editing and supplementation.

Author Response

Manuscript ID plants-2936520

Title: Forgotten Gems: Exploring the Untapped Benefits of Underutilized Legumes in Agriculture, Nutrition, and Environmental Sustainability

Dear Reviewer we thank you for your valuable work and the suggestions. The comments have been handled and a point-by-point explanation has been provided.

Reviewer 2

The article is interesting and contains important information about sustainable agriculture and nutrition. Its main weakness is the lack of specific examples or case studies that discuss the concepts presented.

The 'Introduction' completely lacks a clear definition of "underutilized legumes" and examples of such species. The weaknesses of these plants, such as low yields, agrotechnical problems during cultivation and low profitability of cultivation, were also completely ignored.

Response: A better definition of underutilized legumes as well as examples of such legumes have been provided. Also, the limitations in the adoption of the legumes have been provided.

Chapter 1.1 should have the shortened title: Benefits of underutilized legumes in agriculture, nutrition, and sustainability.

Response: This title has been shortened

Line 79-80

I do not agree with the statement that legumes have lower water needs than many other crops, which is why they can be grown in water-poor regions. Please provide literature data confirming this statement.

Response: The statement has been removed.

In chapter 1.2 (line 104-105) it is written that these plants have unique features, such as resistance to drought and heat, which makes them resistant to climate change, and in chapter 5 (line 381-383) and elsewhere, there is a complete contradiction of this.

Response: The contradiction has been corrected.

The title of chapter 2.1 needs to be edited and shortened. Moreover, the review of "underutilized legumes" is very superficial and covers only 3 species. I suggest supplementing and expanding this chapter.

Response: The sub-title has been shortened.

Similar comments apply to chapters 2.3 and 4.3. They require supplementation and more extensive editing.

Response: More information has been provided and the section has been edited.

Lines 221-223 - I disagree with the statement "Anti-nutrients in underutilized legumes including, α-amylase inhibitors, phytic acids, and lectins have been linked to their ability to resist pests and diseases”. Anti-nutritional substances are also found in popular legumes, which does not make them resistant to diseases.

Response: The information has been removed

The nomenclature should be unified - species names in Latin (see lines 255, 350 et al.)

Response: The nomenclature has been unified.

Table 1 - should contain specific numerical data on the content of individual nutrients. The fact that legumes contain protein and carbohydrates is not a discovery.

Response: numerical data on the amount of individual nutrients has been provided in the Table.

Chapter 3.2 and 3.3 should be combined.

Response: This has been corrected.

Chapter 4.1. is a repetition of what was written earlier. It doesn't show anything new.

Response: This has been removed.

Chapter 4.2. provides very general information. There is a lack of numerical data from the literature on the amount of nitrogen fixation by specific legume species.

Response: The information has been provided.

Chapter 5.1 - the weaknesses of "underutilized legumes" related to low yield potential are presented too briefly and superficially. There is no comparison with other legume species that are popular and widely cultivated.

Response: More information has been provided.

"Conclusions" require extensive editing and supplementation.

Response: This has been effected.

Round 2

Reviewer 1 Report

Comments and Suggestions for Authors

The paper can be published.

Reviewer 2 Report

Comments and Suggestions for Authors

After review the revised version of the manuscript I can see that the Authors have done a lot of work to improve the quality of the article. Undoubtedly, this contributed to achieving the assumed goal - increasing the readability of the article. Most of the suggested corrections were made. The Authors did not supplemented chapter 2.1, so I propose to correct the title to: Overview of selected, underutilized legume varieties.